# The preventive and therapeutic effects of probiotics on mastitis: A systematic review and meta-analysis

**Qinghong Yu[1], Chuchu Xu[1], Mengqian Wang[1], Jiayan Zhu[1], Linghong Yu[1], Zimei Yang[1], Shan Liu[1], Xiufei Gao[1,2]***

**1** First Clinical Medical College, Zhejiang Chinese Medical University, Hangzhou, Zhejiang, China, **2** The First Affiliated Hospital of Zhejiang Chinese Medical University (Zhejiang Provincial Hospital of Chinese Medicine), Hangzhou, Zhejiang, China

* gaoxiufei@zcmu.edu.cn

**Data Availability Statement:** All relevant data are within the paper and its Supporting information files.

**Funding:** This research was supported by The National Natural Science Foundation of China(No.

## Abstract

Acute mastitis is one of the main reasons why breastfeeding women stop breastfeeding, and medication should be used with caution. Considering the uncertainty of mastitis infection and the indications of antibiotic use, as well as the problem of drug resistance and the safety of medication during lactation, probiotics have become an alternative treatment choice. However, a meta-analysis of the effects of probiotics in preventing and treating lactational mastitis is still lacking. Therefore, we searched six electronic databases and the sites of clinical trial registration, a total of six randomized controlled trials were included in this meta-analysis, which showed that oral probiotics during pregnancy can reduce the incidence of mastitis (RR: 0.49, 95% CI: 0.35 to 0.69; p<0.0001). After oral administration of probiotics, the counts of bacteria in the milk of healthy people and mastitis patients were both significantly reduced (in healthy people: MD: -0.19, 95% CI: -0.23 to -0.16, p<0.00001; in mastitis patients: MD: -0.89, 95% CI: -1.34 to -0.43, p = 0.0001). These indicate that to a certain extent, probiotics are beneficial in reducing the incidence rate of mastitis during lactation and some related mastitis symptoms. However, high-quality multicenter clinical trials are still needed to support this result.

## Introduction

Mastitis during lactation is an inflammatory disease, with or without infection [1]. It is common within 4 weeks after delivery, especially in primiparae [2]. Approximately 3% to 15.8% of lactational mastitis cases will develop into breast abscess [3, 4]. Breastfeeding is the best way to feed newborns, but the rate of breastfeeding is far lower than the World Health Organization (WHO) recommendations [5]. The rate of exclusive breastfeeding in the first 6 months is only 40% globally, and up to 33% of puerperae suffer from mastitis and stop breastfeeding [6, 7]. Mastitis not only affects breastfeeding but also affects the mother's physical and mental health. Some authors have proposed that mastitis may be a risk factor for breast cancer, although this assumption has not been fully confirmed [8, 9].

82074438), and the Natural Science Foundation of Zhejiang Province (No. LY18H270006).

**Competing interests:** The authors have declared that no competing interests exist.

It is common to use antibiotics in clinical treatments. However, due to the abuse and resistance of antibiotics, they often lack efficacy, and the use of prophylactic antibiotics after childbirth is also not protective [3]. Furthermore, many mothers are hesitant to take medicine due to its potential exposure to the infant through breast milk metabolism [10]. Therefore, it is necessary to find new effective and safe drugs to replace antibiotics.

Existing evidence has proven that natural breast milk contains its own microbiota, some of which have already been regarded as potential probiotics to help establish the baby's gastrointestinal flora [11, 12]. In recent years, probiotics have been reported to prevent pathogens from colonizing the breast lobules [13]. However, the available evidence suggests that taking probiotics during pregnancy and lactation is safe [14]. However, for the "special periods", they should be used more cautiously [15].

This is the first systematic review and meta-analysis evaluating the preventive and therapeutic effects of probiotics on lactating mastitis. Thus, this review aimed to examine and evaluate the existing evidence for the incidence of lactating mastitis, the improvement of mastitis symptoms and the occurrence of adverse reactions by oral Lactobacillus isolated from breast milk. Robust research is required to provide more options for the prevention and treatment of breastfeeding mastitis.

## Materials and methods

This systematic review has been registered on PROSPERO (registration number: CRD42021257791). For details, please visit: https://www.crd.york.ac.uk/prospero/. We used the Preferred Reporting of Items for Systematic Reviews and Meta-Analyses (PRISMA) [16] as a guideline for reporting this review.

### Search strategy

We conducted this review following the PRISMA guidelines. The following databases were used: PubMed, Web of Science, Cochrane Library, EMbase, China National Knowledge Infrastructure database (CNKI) and Wanfang Data Knowledge Service Platform. Articles published from the day of site construction to November 2021 were searched. The details of the search strategy can be found in S1 Table.

To avoid publication bias, we searched the sites of clinical trial registration sites such as Clinical Trials gov, World Health Organization: International Clinical Trials Registry Platform (WHO ICTRP), the Australian New Zealand Clinical Trials Registry (ANZCTR), and International Standard Randomized Controlled Trial Number Register (ISRCTN). Reference lists were also included, and no language or date restrictions were applied.

We have a professional who conducts regular searches on the database, namely, weekly searches of the above databases and monthly searches of the above clinical trials registry.

### Eligibility criteria

There are four basic criteria that the selected articles needed to satisfy: (1) inclusion of patients with mastitis before or after interventions; (2) design was a randomized controlled trial (RCT) or quasi-RCT; (3) therapy description included the probiotic strains, doses and intervention period; and (4) outcomes and curative effect were reported.

Articles were excluded if they met the following criteria: (1) full text was not available; (2) the outcome was not of interest; (3) research results were not public; and (4) the article types were publications such as discussion papers, letters, reviews and conference reports.

The articles were selected independently by two researchers. When there were differences of opinion related to any article, we reached consensus through discussion. If in doubt, we also analyzed the content of the full text.

### Data and outcome extraction and quality assessment

The secondary researcher independently extracted the characteristic data of the included studies (e.g., the first author's name and publication year, sample size, age of participants, probiotic strain and dose, and period of intervention) and verified them with the primary researcher. The predetermined primary outcome was the incidence of lactating mastitis and the occurrence of breast pain, as well as the bacterial count in milk. The secondary outcome was the level of interleukin-8 (IL-8) in breast milk and the adverse reactions after oral probiotics.

We used the tools of the Cochrane collaboration [17] to assess the risk of bias in all included studies and resolved any disagreements through discussion. Each evaluated item was classified as low risk (+), high risk (-), or unclear (?). All data extraction and quality evaluation were conducted by two researchers individually, and disagreements were resolved by discussion to reach a consensus.

### Statistical analysis

Review Manager (RevMan 5.3) was used for statistical analysis. We applied risk ratios (RRs) and 95% CIs to report the results of dichotomous data and mean differences (MDs) and 95% CIs to report the results of continuous data. Among them, heterogeneity was evaluated by the Q statistic and $I^2$ statistic. When $I^2$ was greater than 50%, the random effects model was used for statistical analysis, and the origin of heterogeneity was analyzed by sensitivity or subgroup analyses. The relevant data are represented by a forest map. When the observation indices could not be combined, we used descriptive language to express the results. Due to the small number of studies finally included, we did not perform publication bias analysis.

## Results

### Study selection

A total of 807 records were initially searched, and 512 retrieved publications were screened by title and abstract for the first selection after deleting duplicates. Among them, 489 records were excluded for various reasons. A total of 23 records were screened for the second time through a full-text review. Finally, 6 studies met the inclusion criteria and were included in this review [18–23]. A PRISMA flow diagram of the search summary identification and selections of the studies is shown in Fig 1.

### Risk of bias (Quality) assessment

The risk of bias in individual studies is shown in Figs 2 and 3. Since 1 study [18] did not provide a specific randomization process, we judged it as high risk. One study [22] did not report all results according to the experimental protocol, which was considered to have a high risk of reporting bias. Another study [21] was considered to be at high risk of attrition bias and reporting bias because it did not report all the outcomes fully.

### Study characteristics

In this review, a total of six studies including 1197 participants assessed the preventive or therapeutic effects of probiotics on mastitis. In three of the studies [21–23] including 727 participants, the intervention was initiated in the prenatal period. Only Jiménez et al. [20] used a

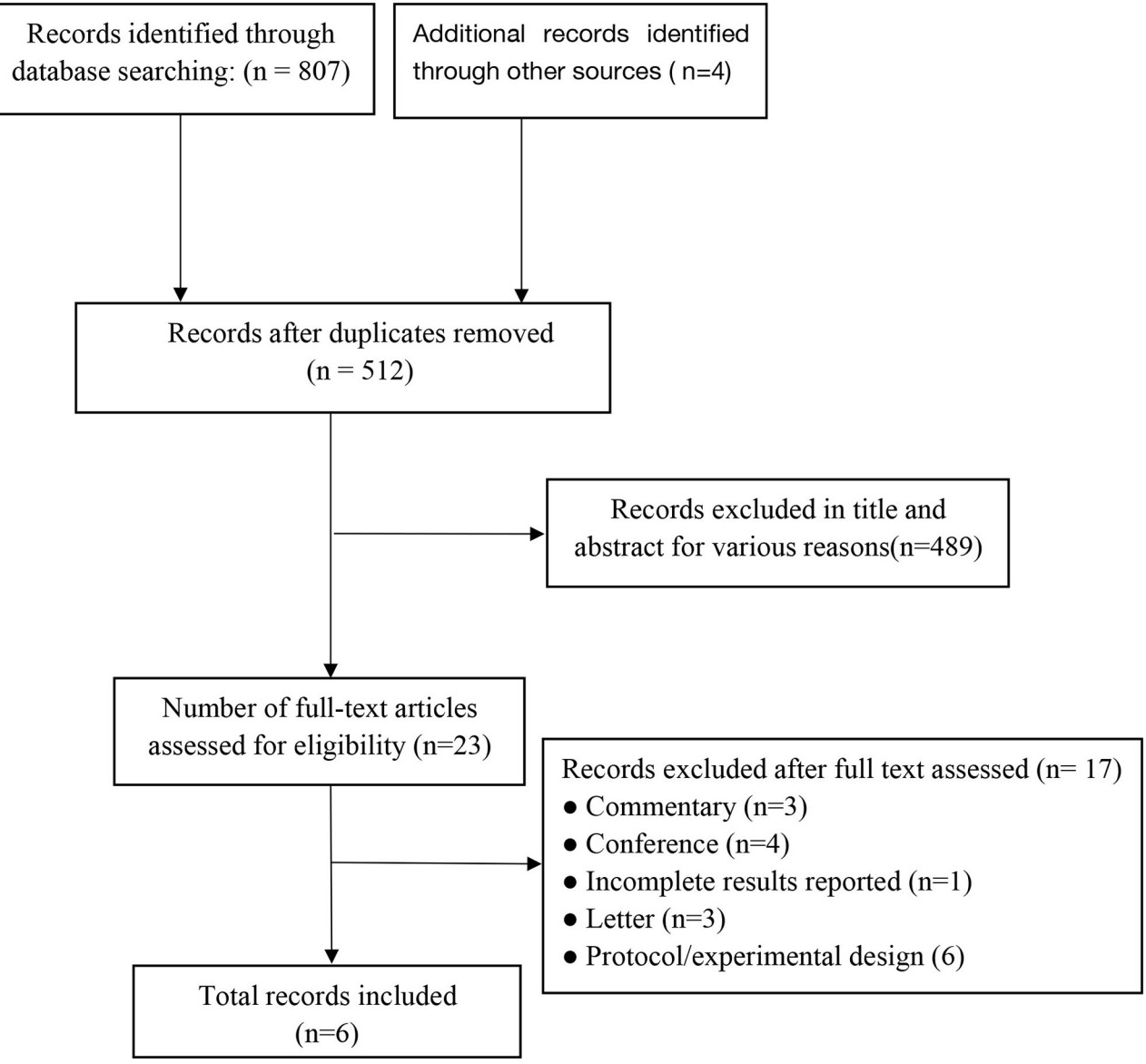

**Fig 1. PRISMA flow diagram describing the process of record identification and selection.**

mixture of two types of probiotic strains for the intervention. Arroyo et al. [18] separately compared the efficacy of the two probiotic strains and antibiotics, and Maldonado-Lobón et al. [19] compared different doses of probiotics. As of the writing of this review, only Arroyo et al. [18] compared the effects of probiotics and antibiotics in the treatment of breastfeeding-related mastitis. All the studies were published from 2008 to 2021 and published in English. The corresponding information is outlined in the tables (Table 1). For the 4 excluded randomized trials that are still in the recruitment stage or have unpublished results, we summarized the information in the form of tables in the supplement (S2 Table).

## Outcomes

**Incidence of mastitis.**   In this meta-analysis, only 3 studies reported the incidence of mastitis. The incidence of mastitis in women taking probiotics was significantly lower than that in women taking the placebo (RR: 0.49, 95% CI: 0.35 to 0.69; $I^2$ = 0%, p<0.0001) (Fig 4).

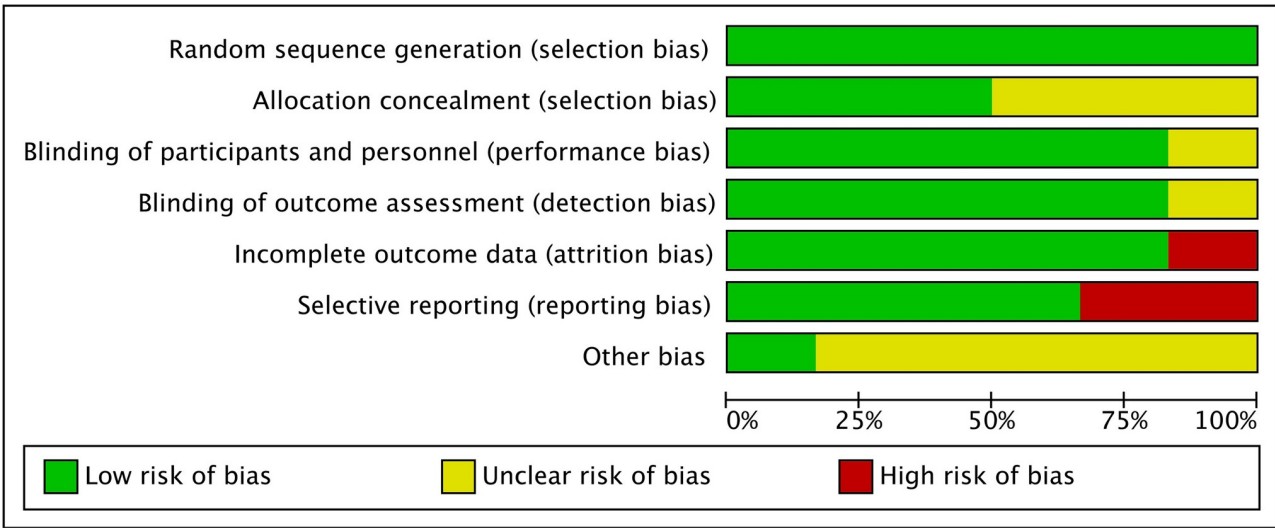

**Fig 2. Risk of bias graph: Review authors' judgments about each risk of bias item presented as percentages across all included studies.**

**Breast pain.** Due to the differences in statistical methods used, the outcomes of some studies cannot be combined. Only 2 studies [21, 22] were included in this outcome indicator. For those with mastitis, women taking probiotics had a lower incidence of breast pain, although the analysis did not reach statistical significance (RR: 0.81, 95% CI: 0.64 to 1.01; $I^2 =$ 50%, p = 0.07) (Fig 5).

**Bacteria count.** The 5 included studies [18–22] reported people with mastitis taking probiotics, and the bacterial count in milk was significantly less than that in the control group (MD: -0.89, 95% CI: -1.34 to -0.43; $I^2 = 93\%$, p = 0.0001) (Fig 6). Similarly, this difference was observed in healthy participants after taking probiotics in 2 studies (MD: -0.19, 95% CI: -0.23 to -0.16; $I^2 = 0\%$, p<0.00001) (Fig 7).

Due to the high heterogeneity in the mastitis group, we used sensitivity analysis by the leave-one-out method, and the result was stable. However, even if high-risk studies were excluded, heterogeneity could not be eliminated. We performed subgroup analysis according to the probiotic strains and intervention doses, and the heterogeneity decreased. The use of Lactobacillus fermentum (MD: -0.55, 95% CI: -0.73 to -0.37; $I^2 = 8\%$, p<0.00001) or Lactobacillus salivarius (MD: -0.90, 95% CI: -1.05 to -0.75; $I^2 = 0\%$, p<0.00001) effectively reduced the bacterial load in milk (Fig 8).

**IL-8.** Two studies [19, 21] showed that the consumption of probiotics caused a decrease in the level of IL-8 in milk (MD: -0.38, 95% CI: -0.58 to -0.19; $I^2 = 0\%$, p<0.0001) (S1 Fig).

**Adverse reactions.** Only 3 studies [18, 19, 23] reported adverse results. During the 3 months of follow-up by Arroyo et al., [18] there were 9 women describing flatulence after taking antibiotics and 9 women suffering from vaginal candidiasis after oral probiotics. Jiménez et al. expressed that all adverse reactions were unrelated to the intake of probiotics. Maldonado-Lobón et al. [19] did not provide data on adverse reactions but only used descriptive language to express that there was no significant difference in diarrhea and flatulence between the probiotic and placebo groups. We contacted the author for detailed data; regretfully, we did not receive a reply. Thus, the data in the original article could not be combined. Two studies reported that oral probiotics can reduce the recurrence of mastitis (RR: 0.29, 95% CI: 0.18 to 0.48; $I^2 = 0\%$, p<0.00001) (S2 Fig).

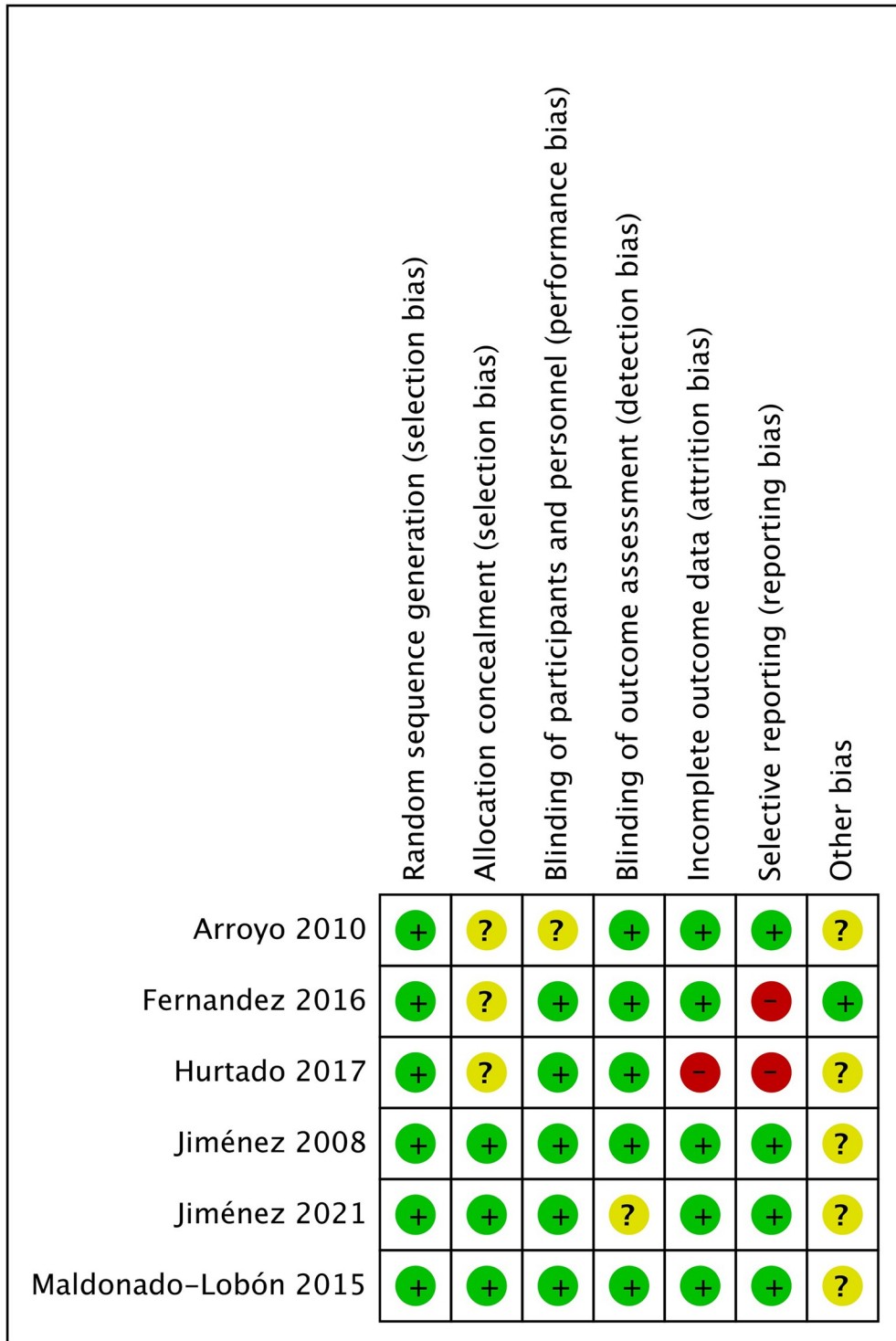

**Fig 3. Risk of bias summary: Review authors' judgments about each risk of bias item for all included studies.**

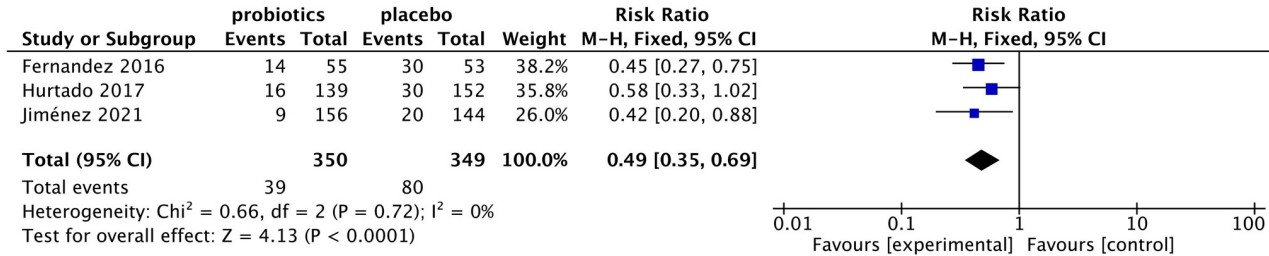

**Fig 4. Forest plot of the incidence of lactation mastitis.**

## Discussion

The use of antibiotics in the treatment of lactating mastitis has always been a controversial topic. Currently, the emergence of multiple drug resistance of various pathogenic bacteria to antibiotics increases the difficulty of disease cure and leads to mastitis more easily. The occurrence of mastitis afflicted the mother, resulting in weaning and changes in the nutritional

**Table 1. Summary of the characteristics of the included studies.**

| First author [ref.] and year | Participant characteristics | Sample size (N) EG/CG | Age (years) EG/CG | Probiotic strain | Doses and administration pattern | Period of invention (weeks) | *Outcome indicators |
|---|---|---|---|---|---|---|---|
| **Jiménez et al. 2008** | Milk bacterial count >4 log10 CFU/mL, and milk leukocyte count >6 log10 cells/mL; Mammary abscess or any other mammary pathology was excluded | 10/10 | NR | Lactobacillus salivarius CECT5713 and Lactobacillus gasseri CECT5714 | $1\times10^{10}$CFU/day | ~4 | 3,6 |
| **Arroyo et al. 2010** | | 124(A group) 127(B group) /101(C group) | NR | Lactobacillus fermentum CECT 5716/Lactobacillus salivarius CECT5713 | $1\times10^{9}$CFU/day | 3 | 2,3,4,6 |
| **Maldonado-Lobón et al. 2015** | milk bacterial count >3 log10 CFU/mL; Mammary abscess or any other mammary pathology was excluded | 23(EG I) 24(EG II) 24(EG III) 27(CG) | 33.3±5.2 34.3±4.3 36.0±2.8 33.4±4.5 | Lactobacillus fermentum CECT 5716 | $3\times10^{9}$(EG)CFU/day $6\times10^{9}$(EGI)CFU/day $9\times10^{9}$(EGII)CFU/day | 3 | 2,3,4,5 |
| **Fernández et al. 2016** | Healthy pregnant women who had not taken probiotic supplements or received antibiotic treatment in the past 30 days | 55/53 | 31.18 (±0.48)/ 30.51 (±0.49) | Lactobacillus salivarius PS2 | $1\times10^{9}$CFU/day | ~8 | 1,2,3,6 |
| **Hurtado et al. 2017** | Healthy pregnant women who have received prophylactic antibiotics 48 hours before and after delivery | 139/152 | 31.91 (±0.49)/ 32.19 (±0.48) | Lactobacillus fermentum | $3\times10^{9}$CFU/day | 16 | 1,2,3,5 |
| **Jiménez et al. 2021** | Healthy pregnant women ≥18 years of age and with the intention to breastfeed | 165/163 | 33.00 (±3.00)/ 33.00 (±3.00) | Lactobacillus salivarius PS2 | $1\times10^{9}$CFU/day | from the 35th week of pregnancy until week 12 after delivery | 1,2,4 |

N: the number of subjects; EG: experimental group; CG: control group; NR: not reported; N: cannot be detected

*Outcome indicators: 1 = incidence of mastitis; 2 = breast pain; 3 = bacteria count in milk samples; 4 = adverse reaction; 5 = immune parameters in breast milk;

6 = detection of the lactobacillus strains in milk samples

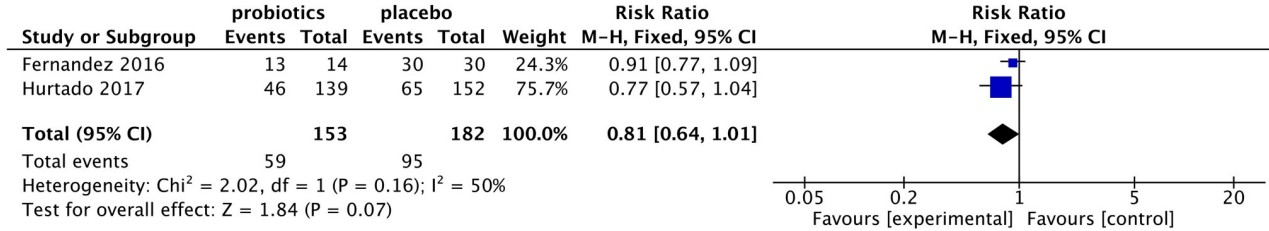

**Fig 5. Forest plot of the incidence of breast pain.**

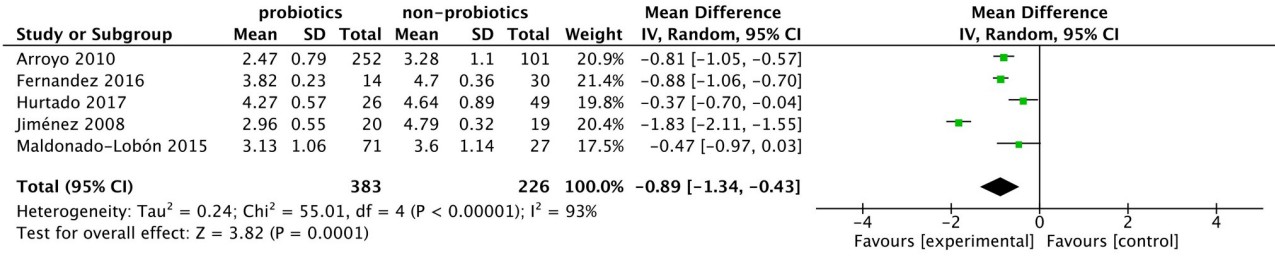

**Fig 6. Forest plot of bacterial counts in patients with lactational mastitis.**

intake of the baby. The animal husbandry and dairy industries also face major challenges due to mastitis of cattle and sheep, which greatly reduces the economic benefit. Therefore, it is urgent to find new prevention and treatment methods.

We found that probiotics had an effect on reducing the incidence of mastitis and improving the symptoms of breastfeeding mastitis according to the currently published clinical trial data. Although the number and quality of the published experiments are not sufficient, there is still a lack of long-term follow-up of volunteers to evaluate long-term efficacy and adverse reactions. This was consistent with some of the results in a review [24] in the Cochran library. The author summarized the RCTs of postpartum treatment measures to prevent breastfeeding mastitis and noted that probiotics may reduce the risk of mastitis more than placebo; however, due to the low reliability of the current evidence, it was not possible to conclude that probiotics could reduce the risk of breast pain and breast injury.

Regarding the bacterial count in lactation mastitis, the included related studies [19, 22] indicate that the pain of mastitis is positively correlated with the bacterial load in the milk, showing an increased abundance of opportunistic pathogens (especially enrichment of aerobic bacteria), and consumption of symbiotic anaerobes can easily cause mastitis [25, 26]. However,

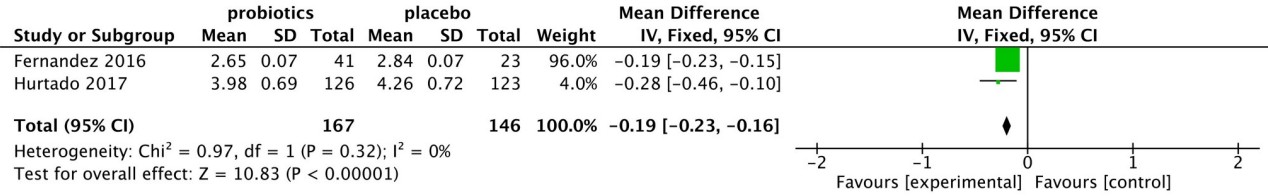

**Fig 7. Forest plot of bacterial counts in healthy women.**

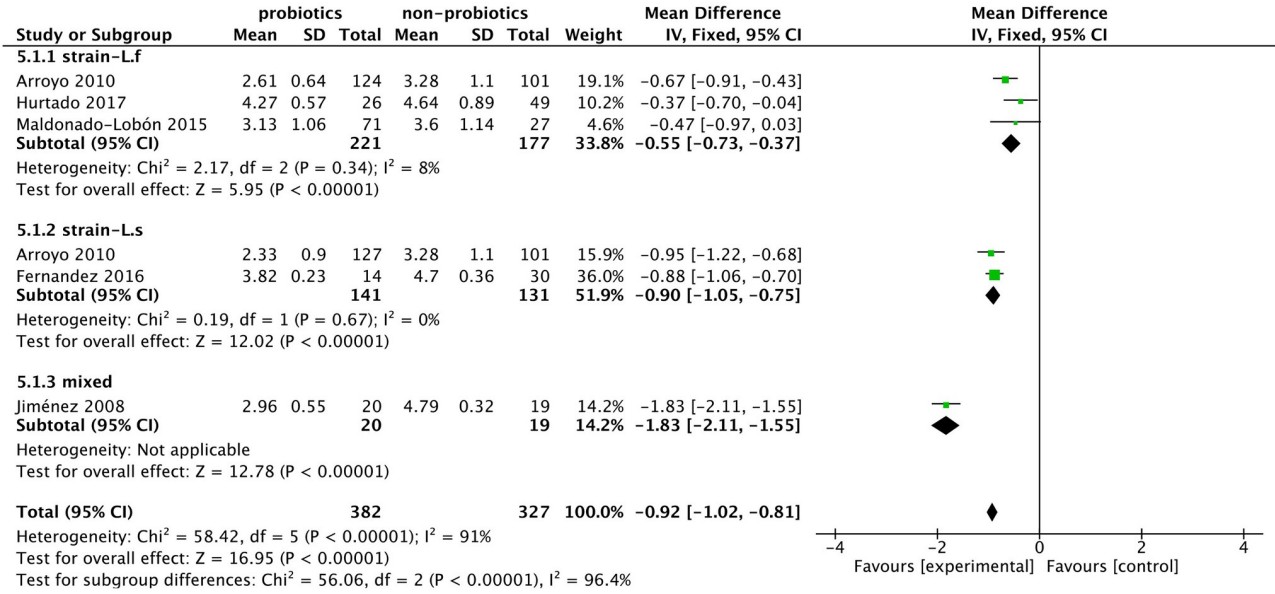

**Fig 8. Forest plot of the subgroup analysis of bacterial counts in mastitis patients.**

some researchers believe that the load of colonies of pathogens in milk has nothing to do with the severity of mastitis symptoms, and numerous infectious pathogens can also be detected in the healthy milk of lactating women [27]. This result demonstrates that lactation mastitis is closely related to the dysbiosis of breast milk's own flora. Hurtado et al. [21] detected that with the increase in the content of IL-8 in milk, the probability of suffering from mastitis increased by fivefold. However, in the clinic, the microbiological, biochemical, or immune biomarkers that indicate the effects of probiotics are still inconclusive. Espinosa-Martos [28] indicated that the intake of Lactobacillus salivarius PS2 leads to a decrease in the level of IL-8 in milk, an increase in immunoglobulins, such as IgG, and increases in the levels of epidermal growth factor (EGF) and transforming growth factor-β (TGF-β2) in human milk, which aid the immune regulation of the breast epithelium. These potential markers may help determine the effects of probiotics in future clinical studies.

Among the excluded studies, there were 4 ongoing trials including over 1,400 participants, which would greatly increase the evidence that probiotics prevent or treat the occurrence of mastitis. Bond D et al. (ACTRN12615000923561) [29] recruited 620 healthy women with a pregnancy greater than 37 weeks and randomly distributed them into two groups. A total of 311 of them were given L. fermentum CECT5716 at 1×1010 CFU per day, and 309 received a placebo to evaluate the effectiveness of oral probiotics in preventing breastfeeding mastitis. Regrettably, the experimental data have not been fully published to date. Karlsson et al. [30] published a cohort study of more than 50,000 women in Norway to examine whether pregnant women taking probiotic milk during pregnancy had a decreased incidence of breastfeeding complications, including mastitis. The results showed that the intake of probiotic milk is positively correlated with the risk of mastitis incidence. However, the authors indicated that this was unlikely to be a causal relationship.

Encouraging results were found in the in vitro assessment of bovine mastitis. The use of Lactobacilli against mastitis pathogens can stimulate an immune response in cows, Staphylococcus aureus biofilm removal, adjustment of the pathogen in a host cell and formation of a biofilm useful against diseases in vitro [31, 32]. Leila et al. [33] showed that Rhamnobacterium

has strong antibacterial activity against Staphylococcus aureus related to infectious or noninfectious mastitis in vitro, with good survival ability under gastrointestinal conditions.

Recently, one of the hypotheses regarding the origin of the breast milk bacterial group, the gut-mammary axis, has brought new ideas for the study of the occurrence and treatment of breastfeeding mastitis. During pregnancy and lactation, the structure and function of the maternal intestines and mammary glands undergo corresponding changes and mutual adaptation [34, 35]. Nonpathogenic intestinal-derived bacteria can be transported to the surface of the breast during lactation through an endogenous cellular pathway called monocytes [36]. Among them, dendritic cells (DCs) play a crucial role [37]. By expressing TJ proteins, DCs can directly penetrate tightly bonded intestinal epithelial monolayer cells and directly grasp bacteria without damaging the intestinal mucosa [38]. In the third trimester of pregnancy, physiological and hormonal changes and increased permeability of the intestinal epithelium contribute to the transfer of intestinal bacteria to the mammary gland and the physiological translocation of immune and bacterial cells [12, 36, 39–41]. Related studies [20–22] also proved that the same strain sequence was detected in milk after oral probiotic administration and demonstrated changes in related immune factors after oral probiotic intake [19, 21, 28]. Thus, the manipulation of the maternal gut microbiota by supplemental probiotics will affect milk-related immune factors. Professor Hu and his team [42] observed that mice with an intestinal flora imbalance and mammary gland exposure to Staphylococcus aureus showed changes in mammary tissue morphology and neutrophil infiltration, as well as increased levels of various inflammatory factors. The breaking of the blood milk barrier and intestinal barrier makes it easier for bacteria to enter the systemic circulation, but fecal microbiota transplantation (FMT) can reverse these changes. In an experiment of FMT from mastitis cows to germ-free mice, Ma et al. [43] found that mouse mammary mastitis symptoms appear, and mice fed probiotics with FMT showed significant symptom relief, with a molecular immune response similar to that of healthy mice. This result suggests that dysbiosis of the intestinal microbiota may be a cause of mastitis, and probiotics are an effective strategy for the treatment of mastitis.

There are some limitations of this review that should be acknowledged. First, the number of published studies on the use of probiotics for lactation-related mastitis is small, although we found that relevant clinical trials were registered in the clinical registry. Second, sensitivity analysis proved the stability of the difference in bacterial count in breast milk, but the heterogeneity was very large. Through the subgroup analysis of probiotic strains, we found that different strains and dosages of interventions were the source of heterogeneity in the results. The sample size of some studies was too small, which may impose bias in the research results. However, the existing evidence demonstrates that probiotics are beneficial against various inflammatory and immune diseases, and the continued exploration of the linear relationship between the intestine and the breast is warranted. We cannot ignore the value of probiotics on lactation-related mastitis. Probiotics have provided a new method to prevent and treat acute mastitis, but there is still a lack of systematic and comprehensive multicenter observations, and further research and exploration are needed.

## Conclusion

In conclusion, mastitis is one of the major breastfeeding-related problems in women, and the current effective intervention strategies are still limited. For pregnant women or patients with mastitis, probiotics maybe an alternative intervention that could bring some benefit in preventing the occurrence of mastitis or improving related symptoms. However, there is still a lack of uniformity and scientificity in the selection of probiotic strains and intervention doses, and the diagnostic criteria and efficacy evaluation indicators of lactation mastitis are

inconsistent, all of which are problems. The need for high-quality RCTs on probiotics and mastitis to support this result still remains.

## Supporting information

**S1 Table. Search strategy.**
(DOCX)

**S2 Table. Characters of related ongoing studies.**
(DOCX)

**S3 Table. PRISMA Checklist 2020.**
(DOCX)

**S1 Fig. Forest plot of level of IL-8 in milk.**
(TIF)

**S2 Fig. Forest plot of recurrence.**
(TIF)

## Author Contributions

**Conceptualization:** Qinghong Yu.

**Data curation:** Chuchu Xu, Mengqian Wang, Jiayan Zhu.

**Formal analysis:** Chuchu Xu, Jiayan Zhu.

**Methodology:** Mengqian Wang, Shan Liu.

**Project administration:** Xiufei Gao.

**Supervision:** Zimei Yang, Xiufei Gao.

**Validation:** Linghong Yu.

**Writing – original draft:** Qinghong Yu.

**Writing – review & editing:** Qinghong Yu.

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
