## [Decision Letter · Decision Letter 0]

11 Jun 2022

PONE-D-21-38695The preventive and therapeutic effects of probiotics on mastitis: A Systematic Review and Meta-AnalysisPLOS ONE

Dear Dr. Gao,

Thank you for submitting your manuscript to PLOS ONE. After careful consideration, we feel that it has merit but does not fully meet PLOS ONE’s publication criteria as it currently stands. Therefore, we invite you to submit a revised version of the manuscript that addresses the points raised during the review process. Please submit your revised manuscript by Jul 26 2022 11:59PM. If you will need more time than this to complete your revisions, please reply to this message or contact the journal office at plosone@plos.org. Please include the following items when submitting your revised manuscript:A rebuttal letter that responds to each point raised by the academic editor and reviewer(s). You should upload this letter as a separate file labeled 'Response to Reviewers'.A marked-up copy of your manuscript that highlights changes made to the original version. You should upload this as a separate file labeled 'Revised Manuscript with Track Changes'.An unmarked version of your revised paper without tracked changes. You should upload this as a separate file labeled 'Manuscript'.

We look forward to receiving your revised manuscript.

Kind regards,

Simone Perna, Ph.D

Academic Editor

PLOS ONE

Journal Requirements:

Reviewers' comments:

Reviewer's Responses to Questions

**Comments to the Author**

1. Is the manuscript technically sound, and do the data support the conclusions?

Reviewer #1: Yes

Reviewer #2: Partly

2. Has the statistical analysis been performed appropriately and rigorously? 

Reviewer #1: Yes

Reviewer #2: Yes

3. Have the authors made all data underlying the findings in their manuscript fully available?

Reviewer #1: Yes

Reviewer #2: No

4. Is the manuscript presented in an intelligible fashion and written in standard English?

Reviewer #1: Yes

Reviewer #2: No

5. Review Comments to the Author

Reviewer #1: The preventive and therapeutic effects of probiotics on mastitis: A Systematic Review and Meta-Analysis

Qinghong Yu · Chuchu Xu · Mengqian Wang · Jiayan Zhu · Linghong Yu · Zimei, Yang, Shan Liu and Xiufei Gao*

Yu et al, in this study, meta-analyzed data to identify effects of probiotics on the prevention and treatment of human mastitis. The data of participants of randomized control treatment studies included healthy pregnant and lactating woman taking probiotics systematically by oral route during pregnancy and lactation. Meta-analysis of six data showed that oral doses of prebiotics significantly reduced the incidence of mastitis and tendency of reduction of breast pain rate. Of these, two studies showed highly significant reduction in milk somatic cell count in two studies of healthy woman and five studies of mastitis patients. Increased number of milk somatic cell count is a definite maker of mastitis. It was concluded that, to a certain extent, probiotic intake appears to reduce incidence of mastitis and mastitis-related symptoms in pregnant and lactating woman.

Though, this study explored 807 records, only six studies qualified for inclusion in this review. Due to lack of sufficient data, role of probiotics in reducing mastitis and mastitis associated symptoms are speculative. Though, authors have acknowledged the limitations of study and suggest high-quality multicenter trails to support their results.

Major issues

1. Line 45: A total of six studies included in the meta-analysis. However, later it is mentioned that two studies of healthy people and five studies of mastitis patients, a total of seven studies are discussed. Please explain this discrepancy or make it the statement clearer.

2. Line 46: “……during pregnancy can reduce the incidence of mastitis (RR: 0.49,

95%CI: 0.35 to 0.69; p<0.0001) and breast pain rate (RR: 0.81, 95 %CI: 0.64 to 1.01; p=0.07)”. This statement explains reduction (P < 0.05) in breast pain rate, which is in fact a tendency of reduction in the pain rate (P = 0.07). Please explain or make the necessary changes.

Minor issues

• Please be consistent with using word “Figure” or “Fig.” See the guidelines of PlosOne.

Reviewer #2: The manuscript has included just 6 studies which are not enough to draw a conclusion under the review

The presentation of manuscript is not according to journals instructions/format

The abstract is not in proper format

The Materials and methods and conclusion parts need to be improved

6. PLOS authors have the option to publish the peer review history of their article (what does this mean?). If published, this will include your full peer review and any attached files.

Reviewer #1: **Yes: **Ratan K. Choudhary

Reviewer #2: No

---

## [Author Response · Author response to Decision Letter 0]

9 Jul 2022

Dear editors and reviewers: 

Thank you for your letter and comments on our manuscript titled “The preventive and therapeutic effects of probiotics on mastitis: A Systematic Review and Meta-Analysis” (PONE-D-21-38695). These comments helped us improve our manuscript, and provided important guidance for future research.

We have addressed the editor’s and the reviewers’ comments to the best of our abilities, and revised text to meet the PLOS ONE style requirements. We hope this meets your requirements for a publication.

Since our original manuscript has been professionally polished, there may be some differences in wording from my previous manuscripts, but we did not change the original intention. Relevant revisions made in response to reviewer comments, we marked the revised portions in red and highlighted them yellow in manuscript. The main comments and our specific response are detailed below:

Reviewer 1: 

1. Line 45: A total of six studies included in the meta-analysis. However, later it is mentioned that two studies of healthy people and five studies of mastitis patients, a total of seven studies are discussed. Please explain this discrepancy or make it the statement clearer. 

Response：I am sorry that this part was not clear in the original manuscript, I should explain that during the review process, we included a total of 6 studies, 2 of which included double data of healthy people and patients with mastitis, and the data discussed at the end were all from these 6 studies. My language did not express clearly, the expression has been removed in the revised manuscript.

2. Line 46: “……during pregnancy can reduce the incidence of mastitis (RR: 0.49,

95%CI: 0.35 to 0.69; p<0.0001) and breast pain rate (RR: 0.81, 95 %CI: 0.64 to 1.01; p=0.07)”. This statement explains reduction (P < 0.05) in breast pain rate, which is in fact a tendency of reduction in the pain rate (P = 0.07). Please explain or make the necessary changes.

Response: Thanks for your suggestion, I have revised the abstract accordingly.

3. Please be consistent with using word “Figure” or “Fig.” See the guidelines of PlosOne.

Response: Thanks for your suggestion, I have made the corresponding changes according to the guidlines of PLOS ONE.

Reviewer 2:

1.The manuscript has included just 6 studies which are not enough to draw a conclusion under the review.

Response: Thank you for your suggestion. Due to the widespread abuse of antibiotics and the prudent choice of medication during pregnancy/lactation, our team believes that it is meaningful to evaluate the intervention effect of probiotics at this time. There are really not many relevant studies at present, and we have acknowledged this in the limitation of manuscript. However, since the relevant experiments in the registration have not been published, our research team is also conducting relevant experiments recently. We hope to supplement this evidence when follow-up related research results are available.

2. The presentation of manuscript is not according to journals instructions/format. The abstract is not in proper format.

Response: Thanks for your suggestion. I have already revised the format of our manuscript.

3. The Materials and methods and conclusion parts need to be improved.

Response: Thanks for your suggestion. Since we did not know the specific modification goals, we consulted Liu S, a statistician, again. And made corresponding changes in the materials and methods and conclusion parts, we hope that this revision will be better than before. 

Editors:

Thanks for your suggestion, we have uploaded and all the figure files to the Preflight Analysis and Conversion Engine (PACE) digital diagnostic tool and passed the certification. So, we re-uploaded our figure file with no changes to the data.

We would be glad to respond to any further questions and comments that you may have.

Sincerely yours,

Xiufei Gao

---

## [Decision Letter · Decision Letter 1]

30 Aug 2022

The preventive and therapeutic effects of probiotics on mastitis: A Systematic Review and Meta-Analysis

PONE-D-21-38695R1

Dear Dr. Gao,

We’re pleased to inform you that your manuscript has been judged scientifically suitable for publication and will be formally accepted for publication once it meets all outstanding technical requirements.

Kind regards,

Simone Perna, Ph.D

Academic Editor

PLOS ONE

Additional Editor Comments (optional):

Reviewers' comments:

Reviewer's Responses to Questions

**Comments to the Author**

1. If the authors have adequately addressed your comments raised in a previous round of review and you feel that this manuscript is now acceptable for publication, you may indicate that here to bypass the “Comments to the Author” section, enter your conflict of interest statement in the “Confidential to Editor” section, and submit your "Accept" recommendation.

Reviewer #1: All comments have been addressed

2. Is the manuscript technically sound, and do the data support the conclusions?

Reviewer #1: Yes

3. Has the statistical analysis been performed appropriately and rigorously? 

Reviewer #1: Yes

4. Have the authors made all data underlying the findings in their manuscript fully available?

Reviewer #1: Yes

5. Is the manuscript presented in an intelligible fashion and written in standard English?

Reviewer #1: Yes

6. Review Comments to the Author

Reviewer #1: (No Response)

7. PLOS authors have the option to publish the peer review history of their article (what does this mean?). If published, this will include your full peer review and any attached files.

Reviewer #1: **Yes: **Ratan K Choudhary

---

## [Editor Report · Acceptance letter]

31 Aug 2022

PONE-D-21-38695R1 

The preventive and therapeutic effects of probiotics on mastitis: A Systematic Review and Meta-Analysis 

Dear Dr. Gao:

I'm pleased to inform you that your manuscript has been deemed suitable for publication in PLOS ONE. Congratulations! Your manuscript is now with our production department. 

Kind regards, 

on behalf of

Professor Simone Perna 

Academic Editor

PLOS ONE